# The eSpiro Ventilator: An Open-Source Response to a Worldwide Pandemic

**DOI:** 10.3390/jcm10112336

**Published:** 2021-05-27

**Authors:** Nicolas Terzi, Fabrice Rastello, Christophe Déhan, Marion Roux, Florian Sigaud, Guillaume Rigault, Cyril Fromentin, Adrien Farrugia, Claude Guérin

**Affiliations:** 1University Grenoble Alpes, Inserm, U1042, HP2, 38000 Grenoble, France; 2Medecine Intensive-Réanimation, CHU Grenoble-Alpes, 38000 Grenoble, France; MRoux3@chu-grenoble.fr (M.R.); FSigaud@chu-grenoble.fr (F.S.); grigault@chu-grenoble.fr (G.R.); 3University Grenoble Alpes, INRIA, CNRS, Grenoble INP, LIG, 38000 Grenoble, France; frastell@gmail.com; 4MinMaxMedical SAS, 75 v Gabriel Peri, 38400 Saint Martin d’Hères, France; christophe.dehan@minmaxmedical.com; 5FineHeart SAS, Cœur Bersol 28 Avenue Gustave Eiffel, Batiment C, 33600 Pessac, France; cyril.fromentin@fine-heart.com; 6SteadXP SAS, 38000 Grenoble, France; adrienfarrugia@yahoo.fr; 7Medecine Intensive-Réanimation, Groupement Hospitalier Centre, Hôpital Edouard Herriot, Hospices Civils de Lyon, 69366 Lyon, France; claude.guerin@chu-lyon.fr; 8Université de Lyon, 69366 Lyon, France; 9Institut Mondor de Recherches Biomédicales, Inserm 955 CNRS ERL 7000, 94010 Créteil, France

**Keywords:** open-source, mechanical ventilation, pandemic, bench study

## Abstract

Objective: To address the issue of ventilator shortages, our group (eSpiro Network) developed a freely replicable, open-source hardware ventilator. Design: We performed a bench study. Setting: Dedicated research room as part of an ICU affiliated to a university hospital. Subjects: We set the lung model with three conditions of resistance and linear compliance for mimicking different respiratory mechanics of representative intensive care unit (ICU) patients. Interventions: The performance of the device was tested using the ASL5000 lung model. Measurements and Main Results: Twenty-seven conditions were tested. All the measurements fell within the ±10% limits for the tidal volume (V_T_). The volume error was influenced by the mechanical condition (*p* = 5.9 × 10^−15^) and the PEEP level (*P* = 1.1 × 10^−12^) but the clinical significance of this finding is likely meaningless (maximum −34 mL in the error). The PEEP error was not influenced by the mechanical condition (*p* = 0.25). Our experimental results demonstrate that the eSpiro ventilator is reliable to deliver V_T_ and PEEP accurately in various respiratory mechanics conditions. Conclusions: We report a low-cost, easy-to-build ventilator, which is reliable to deliver V_T_ and PEEP in passive invasive mechanical ventilation.

## 1. Introduction

With the current COVID-19 pandemic, both intensive care unit (ICU) beds and ventilators have become resources of utmost value. In some countries, the number of ICU beds was expanded quickly by upgrading step-down units and post-operative recovery rooms. However, the shortage of ventilators became a real concern worldwide [1].

This situation compounded the urgent need to develop ventilator systems that can be rapidly deployed [2]. In situations like a global pandemic [1] and regional emergencies [3], or in low-resource ICUs, a ventilator-sharing strategy that maximizes the number of patients able to receive life-saving treatment has been used [4,5]. To address the issue of ventilator shortages, our group (eSpiro Network) developed a freely replicable, open-source hardware ventilator, which should provide better care for critically ill patients while requiring fewer resources [6].

## 2. Ventilator Description

The process subtending the development of the eSpiro ventilator is a follow-up of the first prototype designed in 2010 by a student from the Massachusetts Institute of Technology (Boston, MA, USA), who had the idea of automatizing the compression of a bag–valve–mask (BVM) Ambu^®^ [7]. Because BVMs are largely deployed in hospitals where healthcare workers maintain oxygenation by squeezing the bag by hand, mechanizing the compression of the bag appears to be a very appealing strategy that satisfies the requirements of a low-cost ventilator and rapid manufacture. Our ventilator follows the same rationale with the objective of being affordable and easy to build.

Our eSpiro prototype uses a similar automatized BVM approach, providing closed loop control, monitoring and safety features, eliminating the need for permanent human operation or attendance (Figure 1). The overall design maximizes ease of sourcing with standard medical components and very few 3D printed compounds. Functionally speaking, the mechanical compression of the bag is performed by two horizontally pivoting arms equipped with 3D printed jaws to maximize the possible drawn volume (>1 L) and to reduce bag wear. The arms are driven by a 4.5 N.m (overkilled) gearless stepper motor, which pulls a paraglider cord winding through a freewheeling pulley on each arm, forming a snatch block reduction system. Millions of breathing cycles were tested, showing no sign of wear. The arms, motor and bag compression structure are made from 20 mm squared aluminum stud bars and connectors. The expiratory valve uses a low-pressure electromagnetic control valve. 

Oxygen is delivered via a reservoir bag, which is a component of the bag–valve–mask. Oxygen reservoir includes two one-way valves. Oxygen flow rate equals, or is higher than, the minute volume of the patient and allows 100% of oxygen to be delivered. Control pressure is derived from the BVM pressure itself for insufflation. Positive end-expiratory pressure (PEEP) is generated by a tube dipping in water at the desired and controlled level. The up and down motion of the tube into the water column is controlled by the stepper motor. The system is completed with a medical device main power supply, a lead backup battery pack, a color light tower and a custom electronic board hosting two pressure sensors. Robust control is achieved by independent software and a state machine running on an STM-32 controller. This controller is connected through a serial port to an independent Raspberry Pi^®^ and its touchscreen display, which also provides data logging. The overall system is hosted in two stacked and attached plastic boxes for a total size of 48 × 40 × 60 cm, weighing 27 kg. This weight was mainly due to the use of two batteries (3.5 Kg each) to deal with power cuts in low-resource countries and the stepper motor. The upper box hosts the BVM operation and the lower one, the electronics and power. The eSpiro ventilator is also equipped with alarms for high and low airway pressure and a touchscreen interface for setting and monitoring airway pressure, respiratory rate, tidal volume (V_T_) and inspiratory/expiratory (I/E) duration ratio. Finally, with the eSpiro, the clinician can manually perform end-expiratory and end-inspiratory airway occlusion to measure total PEEP and plateau pressure, respectively. The ventilator only operates in volume control mode, which is the primary mode used in ICUs in France and during the first week of invasive mechanical ventilation worldwide [8]. The eSpiro furthermore offers the monitoring of tidal volume, plateau pressure and driving pressure. These implementations are in accordance with the recommendations to safely deliver mechanical ventilation, especially to COVID-19 patients [8].

## 3. Evaluation Protocol

To mimic representative ICU patients [9], the performance of the device was tested using the ASL 5000 lung model (Ingmar Medical, Pittsburgh, PA, USA). Standard ventilator circuits (Intersurgical^®^, ntersurgical Ltd., Berkshire, UK) and heat-and-moisture exchangers (HEPA Light Isogard^®^, Gibeck, Gibeck® Iso-Gard HEPA light, Teleflex Inc., Morrisville, NC, USA) were used. The ASL 5000 was set in the passive condition, with three conditions of resistance (set equal during inspiration and expiration) and linear compliance for mimicking normal subjects (normal: compliance 50 mL/cmH_2_O and resistance 10 cmH_2_O/L/s), low compliance/normal resistance to mimic acute respiratory distress syndrome (ARDS) (compliance 40 mL/cmH_2_O and resistance 10 cmH_2_O/L/s) and high compliance/high resistance to mimic chronic obstructive pulmonary disease (COPD) (compliance 60 mL/cmH_2_O and resistance 20 cmH_2_O/L/s).

For each mechanical condition, three V_T_ of 300, 400 and 500 mL, and for each of them three levels of PEEP, 5, 10 and 15 cmH_2_O, were tested, generating 27 different experimental conditions. The eSpiro was set to deliver V_T_ in ambient temperature pressure dry conditions, and the other settings were FiO_2_ 0.21, respiratory rate 20 breaths/min, inspiratory time 0.8 s and constant flow inflation. During each condition, the signals of pressure, flow and volume were recorded as measured by the ASL5000 at a sample rate of 512 Hz. After the recording was launched, a 1 min period was allowed for stabilization and then the next 30–40 consecutive breaths were used for the analysis. In the present study, we report the values of V_T_ and PEEP in each condition. Two metrics were derived: (1) the volume error expressed as a percentage error (measured V_T_ minus set V_T_)/(measured V_T_); (2) the PEEP error expressed as an absolute error (measured PEEP minus set PEEP in cmH_2_O). The V_T_ range was evaluated across the 10% boundaries, which commonly define the accuracy of V_T_ delivery. The values were compared between groups by using an analysis for an unbalanced factorial design. A multiple pairwise comparison using Tukey’s honest significant difference (Tukey’s HSD) method was performed. The statistical analysis was performed using R software version 3.5.2. *p* < 0.05 was considered statistically significant.

## 4. Results

Figure 2A reports the volume error for each of the 27 conditions. The mean error varied from −6.8% (high compliance/high resistance/PEEP15/VT500) to +6.2% (high compliance/high resistance/PEEP5/VT300) and the maximum observed error over all cycles was 8.4% (high compliance/high resistance/PEEP15/VT500). All the measurements fell within the ±10% limits for the V_T_. The volume error was influenced by the mechanical condition (*p* = 5.9 × 10^−15^) at PEEP5 and PEEP15 and the PEEP level (*p* = 1.1 × 10^−12^). However, the clinical significance of these differences is uncertain. Indeed, as shown in Figure 2A, the difference oscillates at the maximum between +19 mL for high compliance/high resistance/PEEP5/VT300 and −34 mL for high compliance/high resistance/PEEP15/VT500.

Figure 2B reports the PEEP error. The mean error varied from 0.0 cmH_2_O (low compliance/normal resistance/PEEP15/VT300) to 0.8 cmH_2_O (high compliance/high resistance/PEEP10/VT400) and the maximum observed error over all cycles was 1.8 cmH_2_O (low compliance/normal resistance/PEEP5/VT400). The PEEP error was not influenced by the mechanical condition (*p* = 0.25).

## 5. Discussion

The huge flow rate of patients with a severe acute respiratory failure admitted to ICUs worldwide during the first wave of the COVID-19 pandemic created a risk of ventilators shortage and urged the healthcare system to quickly implement different strategies to match the demand [1,10]. Strategies of note included splitting ventilators between two patients [11,12], developing easy-to-build, open-source, cheap ventilators [13,14] and using an intermediate ventilator dedicated to emergency rooms and patient transport with an enhanced production at a large scale with the help of non-medical industry. Using the same ventilator for two patients is still an experimental and non-recommended strategy [11]. Jonkman et al. have proposed a gas-powered, patient-responsive automatic resuscitator for use in acute respiratory failure [14]. Even if they demonstrated in a bench and porcine model that the device can be efficient, it requires a caregiver at the bedside. Because the device lacks monitoring displays, challenges arise and monitoring would be necessary to avoid excessive V_T_. Moreover, in contrast to conventional ventilation modes, the pressure cycling mechanism makes the tidal volume delivered directly dependent on the patient’s respiratory mechanics. Garmendia et al. have proposed an easy-to-build non-invasive pressure ventilator [4]. Their ventilator is built using easily available off-the-shelf materials and has a simple design: a high-pressure blower, two pressure transducers and a controller with a digital display and open-source construction details provided for replication. The ventilator was evaluated and compared with a commercially available device at the bench using an actively breathing patient simulator and in 12 healthy volunteers submitted to high airway resistance or to simulated restrictive syndrome.

Contrary to the previous easy-to-build ventilator that used pressure-control ventilation, we deliberately decided to develop a ventilator that allows volume-controlled ventilation. Moreover, at the beginning of the COVID-19 pandemic, the great majority of patients were invasively ventilated and deeply sedated [15]. Thus, we decided to develop a device allowing passive invasive mechanical ventilation.

Our experimental results demonstrate that the eSpiro ventilator is reliable to deliver VT and PEEP accurately in various respiratory mechanics conditions. It is worth noting that the present results regarding the VT are in accordance with previous evaluations of ICU ventilators. The seven different benchmarked ICU ventilators demonstrated an error volume that could reach 10% and the same was true in the study by Garnier et al. [16]. 

We have tested the ventilator for more than 24 h without any modification at a high respiratory rate of 30 cycles/minutes and with a tidal volume of 500 mL using a venTest^®^ (Datrend Systems Inc, Richmond, BC, Canada). The tidal volume at the end of the course was the same. Unfortunately, we did not record the data concerning the error of the tidal volume. This is in progress with the next prototype. However, one of the strengths of this device is that a closed loop feedback ventilation algorithm was designed to adapt to the aging of the BVM, whose mechanical and pneumatic properties do not vary enough after 24 h to prevent it from maintaining ventilation performance. Our study is limited by its bench nature and the lack of comparison with other ICU devices. 

The fact that the ventilators were not tested in vivo limits the implications of the present study in the clinical realm. We did not test several inspired fractions in oxygen (FiO_2_) in this first phase of this proof of concept. Performances of the eSpiro ventilator can be viewed as “acceptable” during the initial phase of respiratory failure. Indeed, at this time, the device is limited to passive ventilation, and there is no ability to respond to spontaneous breathing.

Further development is planned. Future iterations will incorporate changes prompted by the results of our prototype testing. We will implement an inspiratory pressure trigger that allows it to deliver the set tidal volume. With this inspiratory trigger, the ventilator will be able to deliver an assisted control (AC) mode of ventilation. AC mode is one of the most widely used mechanical ventilator modes and is adequate for the management of the majority of most clinical respiratory failure scenarios [17]. The advantage of AC mode is that the patient has an assured minute ventilation to meet physiologic needs for adequate gas exchange. Finally, we will test the ventilator on a porcine model before proposing a real life study in stable patients.

## 6. Conclusions

We presented a low-cost, easy-to-build ventilator, which is reliable to deliver passive invasive mechanical ventilation according to the guidelines. Although it has not been used in France during the COVID-19 pandemic as yet, we expect this innovation to be useful in any future pandemic and for low- or middle-income countries.

## Figures and Tables

**Figure 1 jcm-10-02336-f001:**
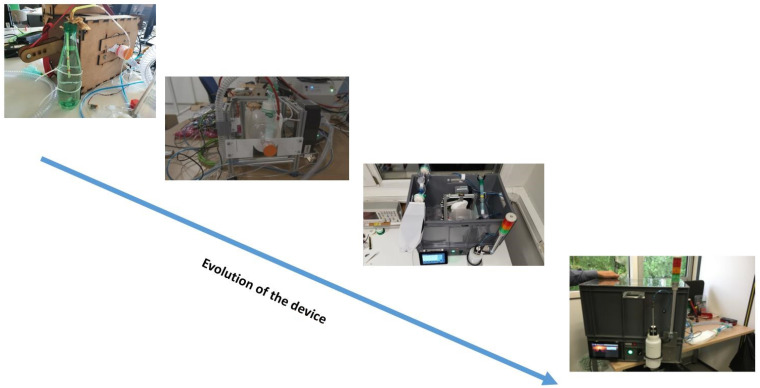
Evolution of the device from the first prototype to the final device.

**Figure 2 jcm-10-02336-f002:**
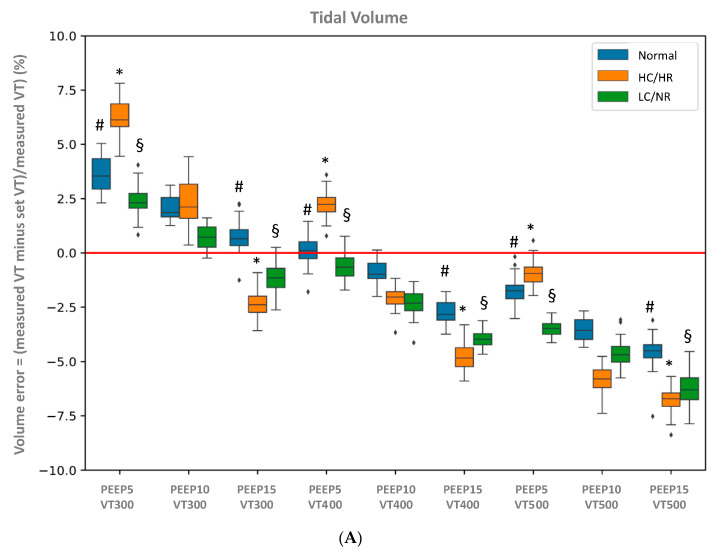
(**A**) Mean values of differences in volume error (% set tidal volume) in each condition (normal, high compliance/high resistance and low compliance/normal resistance). §: Significant difference (*p* < 0.05) in the volume error between high compliance/high resistance and low compliance/normal resistance. *: Significant difference (*p* < 0.05) in the volume error between normal and low compliance/normal resistance. #: Significant difference (*p* < 0.05) in the volume error between normal and high compliance/high resistance. (**B**) Mean values of differences in PEEP error (expressed in cmH_2_O) in each condition (normal, high compliance/high resistance and low compliance/normal resistance). Box-and-Whisker plots of PEEP levels at three nominal PEEP of 5, 10 and 15 cmH_2_O for each tidal volume in each condition. Whiskers denote median ± 1.58 × IQR × √3, where IQR is the interquartile range. The diamonds are the outliers.

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
