# Peer review of "The eSpiro Ventilator: An Open-Source Response to a Worldwide Pandemic"

_jcm, 2021, doi:10.3390/jcm10112336_

Round 1

Reviewer 1 Report

This is an interesting bench test assessment of an easy to develop ventilator system based on an automated Bag Valve Mask system delivering volume control ventilation. The premise is sound and a development of a previous prototype. The indications and needs for such a device in a global pandemic especially in the low and middle income country setting are merit worthy and sound.

The device is well described and the bench testing is simple and clear. There are a couple of major drawbacks which I outline below and the authors already recognise in their brief discussion.

  1. A figure displaying the device in stages of evolution would have been of value for those without a background or detailed knowledge of ventilation.
  2. The bench testing would have been more robust if there was at least one other ventilator asa comparator. This could have been a portable transport device.
  3. For an easy to develop device it seems heavy at a weight of 27KG compared with other comparable size ventilators. I am assuming this is mainly due to the motor required to power the ventilator. Do the authors think this will limit its applicability for what they originally intended its use. This could be brought out in the discussion.
  4. Could the authors describe how they deliver oxygen through the device and have they performed any tests on the error of Fi02 delivered. 
  5. Is there any variation in the error testing when repeated after the ventilator has been running for a long time, e.g at least 24-48 hours to simulate a real life situation.
  6. The conclusion is slightly overstated saying that it performs similarly to high quality commercial devices, as these have not been directly compared in the study. I would not use the term high quality specifically for the commercial devices as this suggests that the device being described is not high quality which should not be the case. It is easy to build and develop and should meet quality standards for build and reliability.
  7. A brief comment on next steps would be useful to conclude the manuscript. 

Author Response

R1

Comments and Suggestions for Authors

This is an interesting bench test assessment of an easy to develop ventilator system based on an automated Bag Valve Mask system delivering volume control ventilation. The premise is sound and a development of a previous prototype. The indications and needs for such a device in a global pandemic especially in the low and middle income country setting are merit worthy and sound.

  1. We are very grateful to the Reviewer for carefully analyzing our manuscript and making constructive comments. Point-by-point replies are provided below.

The device is well described and the bench testing is simple and clear. There are a couple of major drawbacks which I outline below and the authors already recognise in their brief discussion.

  1. A figure displaying the device in stages of evolution would have been of value for those without a background or detailed knowledge of ventilation.

R: Thank you. We have added the evolution from the first prototype to the definitive device.

  1. The bench testing would have been more robust if there was at least one other ventilator as a comparator. This could have been a portable transport device.

R: Thank you for this comment. Unfortunately, we have not compared this new device with a portable transport device. However, we are in progress to compare some innovations launched in France in the field of ventilators that may be used to face a shortage of ICU ventilators.

  1. For an easy to develop device it seems heavy at a weight of 27KG compared with other comparable size ventilators. I am assuming this is mainly due to the motor required to power the ventilator. Do the authors think this will limit its applicability for what they originally intended its use. This could be brought out in the discussion.

R: Thank you for this comment. The weight was mainly due to the 2 batteries, which were inserted. Accordingly, we have notified this point in the “ventilator description” section: “This weight was mainly due to the use of 2 batteries to deal with power cuts in low-resource countries and the stepper motor.” And we also add this point as a limitation in the discussion section of the R2.

  1. Could the authors describe how they deliver oxygen through the device and have they performed any tests on the error of Fi02 delivered.

R: Thank you for this comment. Oxygen is delivered via reservoir bag, which is a component of the bag-valve mask. Oxygen reservoir includes two one-way valves. Oxygen flow rate equals to, or is higher than, the minute volume of the patient and allows 100% oxygen to be delivered. Accordingly, we have added this sentence in the “ventilator description”. Unfortunately, we have not tested different FiO2 during the bench test. We mentioned this as another limitation of present work.

Is there any variation in the error testing when repeated after the ventilator has been running for a long time, e.g at least 24-48 hours to simulate a real life situation.

We have tested the ventilator during more than 24 hours without any modification. Unfortunately, we have not recorded the data concerning the error of the tidal volume. This is in progress with the next prototype. However, the closed loop feedback ventilation algorithm was designed to adapt to the aging of AMBU bags, whose mechanical and pneumatic properties do not vary enough after 24 hours to prevent it from maintaining ventilation performance

We added this in the discussion section. Moreover, as mentioned “Millions of breathing cycles were tested, showing no sign of wear.”

  1. The conclusion is slightly overstated saying that it performs similarly to high quality commercial devices, as these have not been directly compared in the study. I would not use the term high quality specifically for the commercial devices as this suggests that the device being described is not high quality which should not be the case. It is easy to build and develop and should meet quality standards for build and reliability.

Thank you for this comment. We modified the conclusion accordingly as: “that allows passive invasive mechanical ventilation according to the guidelines”

  1. A brief comment on next steps would be useful to conclude the manuscript.

Thank you for this comment. Accordingly, we added these information: “Further development of this proof-of-concept is planned. Future iterations will incorporate changes prompted by the results of our prototype testing. We will implement an inspiratory pressure trigger that allows to deliver the set tidal volume. With this inspiratory trigger added the ventilator will be able to deliver assisted-control (AC) mode of ventilation. AC mode is one of the most widely utilized mechanical ventilator modes, is adequate for the management of  of most clinical respiratory failure scenarios. The advantage of AC mode is that the patient has an assured minute ventilation to meet physiologic needs for adequate gas exchange. Finally, we will test the ventilator on a porcine model before proposing a real life using in stable patients.

Reviewer 2 Report

This study investigated the performance of a simple ventilator developed by the authors in response to the shortage of ventilators that was a problem during the pandemic. Overall, this is a well-written paper.

Major Comments

Discussion is not enough. Other similar simple ventilators have been proposed, and a multifaceted comparison with them needs to be made.

Limitation needs to be clearly written. We believe that the use of the system will be limited due to the inability to respond to spontaneous breathing.

Minor Comments

I think the conclusion of the abstract is overstatement. We believe that some accuracy such as PEEP and VT does not make it similar to a commercial high-performance ventilator.

P1 Line 40- reference was needed.

It would be easier for readers to understand if there were pictures and illustrations.

Author Response

R2

This study investigated the performance of a simple ventilator developed by the authors in response to the shortage of ventilators that was a problem during the pandemic. Overall, this is a well-written paper.

Major Comments

Discussion is not enough. Other similar simple ventilators have been proposed, and a multifaceted comparison with them needs to be made.

R :  Thank you for this comment. Accordingly, we modified the discussion section.

Limitation needs to be clearly written. We believe that the use of the system will be limited due to the inability to respond to spontaneous breathing.

R :  Thank you for this comment. Accordingly, we modified the limitation section.

Minor Comments

I think the conclusion of the abstract is overstatement. We believe that some accuracy such as PEEP and VT does not make it similar to a commercial high-performance ventilator.

R :  Thank you for this comment Accordingly, we modified the conclusion: “We presented a low-cost, easy to build ventilator which is reliable to deliver passive invasive mechanical ventilation.

P1 Line 40- reference was needed.

R :  Thank you for this comment. Added as requested.

It would be easier for readers to understand if there were pictures and illustrations.

R :  Thank you for this comment. Added as requested.

Reviewer 3 Report

Over all a nice bench study. But it is a bench study, so wording like “Setting: ICU of Grenoble Teaching Hospital, France” in lane21 seems not correct“. The performance of the device were tested“ in lane 24 seems also an overinterpretation as basically only VT and PEEP were tested and this only in short time situations and not in a longer period of test. Again: “that performs similarly to high-quality commercial devices“ lane 32 and 33 is obviously not correct as a high-quality device may or at least should have more features than just applying VT and PEEP. Please change the multiple passages with this not correct or “overinterpretations”.

I would like to see a picture of the device as the formal description is not easy to understand.

Lane 93 -98: “The ASL 5000 was set in passive condition, with three conditions of resistance (set equal in inspiration and expiration) and linear compli-ance for mimicking normal subjects (Normal: Compliance 50 ml/cmH2O and Resistance 10 cmH2O/L/s), acute respiratory distress syndrome (ARDS) (Compliance 40 ml/cmH2O and Resistance 10 cmH2O/L/s), and chronic obstructive pulmonary disease (COPD) (Compliance 60 ml/cmH2O and Resistance 20 cmH2O/L/s).” Indeed the ASL mimicking clinical conditions. Of course a model can not simulate ARDS or COPD as these a clinical conditions/syndromes. I would recommend to change ARDS to low compliance / normal resistance and COPD to high compliance/hight resistance throughout the manuscript especially in the figures.

Lane 118; “The vol-ume error was influenced by the mechanical condition (P=5.9 e -15) and the PEEP level (P=1.1e -12)” Ok. Is this important? If so please describe further the error. If not please delete.

Limitations: please include that the device was only tested in passive condition of the ASL 5000

In view of the brevity of the manuscript and the simple experimental and easily manageable structure of setting I would recommend to publish the manuscript as a short communication rather than an article

Minor:

It looks like the automatic hyphenation was activated.

H2O should be H2O

Author Response

R3

Over all a nice bench study. But it is a bench study, so wording like “Setting: ICU of Grenoble Teaching Hospital, France” in lane21 seems not correct“.

  1. Thank you for this comment. In the R2 we have modified as follows: Setting: Dedicated research room taking part of an ICU affiliated to a University Hospital.

The performance of the device were tested “in lane 24 seems also an overinterpretation as basically only VT and PEEP were tested and this only in short time situations and not in a longer period of test. Again: “that performs similarly to high-quality commercial devices“ lane 32 and 33 is obviously not correct as a high-quality device may or at least should have more features than just applying VT and PEEP. Please change the multiple passages with this not correct or “overinterpretations”.

R :  Thank you for this comment. Modified as suggested by the reviewers 1 and 2.

I would like to see a picture of the device as the formal description is not easy to understand.

R :  Thank you for this comment. Added as requested.

Lane 93 -98: “The ASL 5000 was set in passive condition, with three conditions of resistance (set equal in inspiration and expiration) and linear compli-ance for mimicking normal subjects (Normal: Compliance 50 ml/cmH2O and Resistance 10 cmH2O/L/s), acute respiratory distress syndrome (ARDS) (Compliance 40 ml/cmH2O and Resistance 10 cmH2O/L/s), and chronic obstructive pulmonary disease (COPD) (Compliance 60 ml/cmH2O and Resistance 20 cmH2O/L/s).” Indeed the ASL mimicking clinical conditions. Of course a model can not simulate ARDS or COPD as these a clinical conditions/syndromes. I would recommend to change ARDS to low compliance / normal resistance and COPD to high compliance/hight resistance throughout the manuscript especially in the figures.

R :  Thank you for this comment. Modified as suggested

Lane 118; “The volume error was influenced by the mechanical condition (P=5.9 e -15) and the PEEP level (P=1.1e -12)” Ok. Is this important? If so please describe further the error. If not please delete.

R: Thank you for this comment. You are right in the sense that the statistical significance resulted from the specific bench conditions with a very high between-breaths reproducibility and a large number of breaths analyzed. Therefore, the clinical significance of this result is unclear and hence, we deleted this part as you suggested.

Limitations: please include that the device was only tested in passive condition of the ASL 5000

R :  Thank you for this comment. Modified as suggested

In view of the brevity of the manuscript and the simple experimental and easily manageable structure of setting I would recommend to publish the manuscript as a short communication rather than an article

Minor:

It looks like the automatic hyphenation was activated.

H2O should be H2O

  1. Thank you for this comment. This typo was fixed in the R2.